# Advancing Leukemia Management Through Liquid Biopsy: Insights into Biomarkers and Clinical Utility

**DOI:** 10.3390/cancers17091438

**Published:** 2025-04-25

**Authors:** Cíntia Nogueira Hollanda, Ana Cristina Moura Gualberto, Andréa Barretto Motoyama, Fabio Pittella-Silva

**Affiliations:** Laboratory of Molecular Pathology of Cancer, Faculty of Health Sciences, University of Brasilia, Brasilia 70910-900, Brazil; cintia.hollanda@aluno.unb.br (C.N.H.); ana.gualberto@unb.br (A.C.M.G.); andreabm@unb.br (A.B.M.)

**Keywords:** liquid biopsy, biomarkers, acute lymphoid leukemia, chronic lymphoid leukemia, acute myeloid leukemia, chronic myeloid leukemia, minimal residual disease (MRD), treatment response

## Abstract

Leukemia is a diverse group of hematological malignancies that have their origin in the bone marrow. Conventional methods for disease monitoring frequently depend on invasive procedures, which can impose constraints on continuous evaluation. This study explores the potential of liquid biopsy as a minimally invasive strategy for the detection and monitoring of leukemia through the analysis of blood-derived biomarkers. These include cell-free DNA, circulating tumor DNA, circulating tumor cells, microRNAs, and extracellular vesicles, which are released by tumor cells and may reflect disease burden and therapeutic response. This review aims to examine how liquid biopsy can enhance clinical decision making by enabling the early detection of disease relapse, tracking clonal evolution, and assessing treatment efficacy. The results have the potential to enhance the clinical applicability of liquid biopsy in the management of leukemia and to promote its expanded integration into precision oncology strategies.

## 1. Introduction

Leukemias are hematological malignancies characterized by the uncontrolled proliferation of dysfunctional clonal cells from one or more hematopoietic lineages. This abnormal proliferation leads to the accumulation of malignant cells in the bone marrow, resulting in impaired hematopoiesis and reduced production of functional blood cells and platelets. Additionally, the extramedullary accumulation of these dysfunctional cells can manifest as lymphadenopathy, hepatomegaly, splenomegaly, or central nervous system (CNS) involvement [1,2].

This group of diseases is classified based on the affected lineage and the rate of cellular proliferation. The most encountered subtypes in clinical practice include acute myeloid leukemia (AML), acute lymphoblastic leukemia (ALL), chronic myeloid leukemia (CML), and chronic lymphocytic leukemia (CLL) [2].

The diagnosis and monitoring of leukemias rely on classical laboratory methods such as complete blood count, cell morphology analysis, and immunophenotyping, supplemented by bone marrow aspiration and biopsy [1,2,3,4]. Although bone marrow biopsy remains the gold standard for diagnosis, it presents significant limitations. A single aspiration may fail to capture the genetic heterogeneity of the tumor microenvironment, and the procedure is both invasive and risky, being impractical for repeated use [5].

In response to these challenges, innovative methods have been developed to enable non-invasive analysis of tumor burden and genetic alterations, thereby supporting the diagnosis, monitoring, and prognosis of hematological malignancies.

Liquid biopsy is an emerging technique that traditionally involves the analysis of liquid samples from bodily fluids (such as blood, urine, saliva) to detect and characterize small fragments of circulating cell-free DNA (cfDNA), which are released during apoptosis by both healthy and malignant cells [6]. In addition to ctDNA, other circulating biomarkers, such as extracellular vesicles, microRNAs (miRNAs), and circulating tumor cells (CTCs) are also explored for their potential utility in liquid biopsy [7].

This comprehensive review aims to explore the clinical significance and usage of liquid biopsy for diagnosis, prognosis, and monitoring of treatment responses in lymphoid and myeloid leukemias.

### 1.1. Liquid Biopsy Analytes

Liquid biopsy techniques used in oncology examine several analytes, including circulating tumor cells (CTCs), circulating tumor DNA (ctDNA), circulating tumor RNA (ctRNA), extracellular vesicles, and metabolites.

Circulating tumor cells (CTCs) are cells shed from the primary tumor into circulation that may be detected through label-dependent and label-independent methods. The clinical utility of CTCs has been explored in different solid tumors, most specifically breast, prostate, colorectal, and non-small- and small-cell lung cancer [8].

Circulating tumor RNA (ctRNA) is a broad term referring to several types of RNA, encompassing mRNAs, long non-coding RNAs (lnRNAs), and small non-coding RNAs (snRNAs)—the last including microRNAs (miRNAs) and circular RNAs (circRNAs) [9]. ctRNAs are released passively through cell death or actively as part of extracellular vesicles or lipoprotein complexes and have been suggested as promising biomarkers in cancer [10]. When actively secreted, they may help regulate gene expression and cell function at distant sites [11].

MicroRNAs (miRNAs) are small non-coding RNAs that regulate gene expression through binding to target messenger RNAs (mRNAs). These molecules have an important role in oncogenesis, being able to behave as oncogenes or tumor suppressors [11], depending on the function of their silenced target. Extracellular miRNAs are also considered attractive biomarkers due to their stability in bodily fluids and general resistance to damage [12]. miRNAs have been explored as diagnostic and prognostic markers in solid tumors such as breast, colorectal, gastric, lung, prostate, ovarian, and pancreatic cancer, as well as glioblastoma, hepatocellular carcinoma, and melanoma [11]. In leukemia, several studies have identified dysregulated miRNA expression in ALL and AML and, less frequently, in CLL and CML [12,13]. In 2021, Zhang et al. [13] conducted a meta-analysis of 49 studies with over 3400 patients and 2700 healthy controls. Those authors concluded that circulating miRNAs could distinguish leukemia patients from healthy controls with high diagnostic accuracy (overall sensitivity and specificity were 0.83, 0.92, respectively) [13].

Extracellular vesicles (EVs) can be defined as cell-released particles delimited by a lipid bilayer that do not have the ability to self-replicate. There are three main subtypes of EVs: microvesicles (MVs), exosomes, and apoptotic bodies. They are classified according to their biogenesis, release pathways, size, density, content, and function. Although their content may vary, it includes lipids, nucleic acids, and proteins—specifically, those associated with the plasma membrane, cytosol, and lipid metabolism [14]. Exosomes are small-scale extracellular vesicles, being around 30–200 nm in diameter [15]. EVs are released continuously from both normal and tumor cells, and because some of them may fuse with the membranes of cells, they have a known role in cancer initiation and progression. Their potential as liquid biopsy biomarkers have been explored in both solid and hematological cancers due to their content profile of DNA, RNA, and proteins mirroring their cell of origin [15,16].

Altered metabolism in cancer cells leads to the production of aberrant metabolites with potential utility as biomarkers for cancer pathogenesis and development. Circulating metabolites have been studied as diagnostic biomarkers in solid tumors such as prostate, bladder, and pancreatic cancer, but discrepancies such as lack of uniform detection methodologies and metabolic heterogeneity still constitute challenges to be overcome [17,18].

### 1.2. Cell-Free DNA and Circulating Tumor DNA

Cell-free DNA (cfDNA) consists of highly fragmented extracellular DNA found in blood, urine, saliva, peritoneal fluid, pleural fluid, uterine lavage fluid, and cerebrospinal fluid [19]. cfDNA may be released through different mechanisms into circulation, being mainly found due to apoptotic and necrotic hematopoietic cells that release DNA during cell destruction [6,20]. While cfDNA is detected in low concentrations in healthy individuals, it is found at an increased level in cancer patients with premalignant states and in inflammation, trauma, after exhaustive exercise, and in elderly patients suffering from acute or chronic illnesses [21]. Circulating tumor DNA (ctDNA) refers to the fraction of cfDNA that originates specifically from cancer cells, and it may account for as little as < 0.01% of the total cfDNA concentration [22]. ctDNA provides information about the genomic landscape of the tumor, and so it is an attractive biomarker for liquid biopsy for early disease detection, identification of cancers of unknown primary site, tumor genotyping, tumor staging and prognosis, guiding treatment selection, monitoring treatment efficacy and resistance, and monitoring minimal residual disease (MRD) [6,7].

Many reviews have highlighted the role of ctDNA as a biomarker in solid tumors including lung, colon, and breast cancer. In lung cancer, clinical practice guidelines have already incorporated the use of liquid biopsy for the detection of the Epidermal Growth Factor Receptor (EGFR) T790M mutation in ctDNA, which will determine treatment follow-up [23]. The amount of ctDNA is known to correlate with tumor stage, tumor burden, and metabolism, being an interesting prognostic marker. Additionally, it is also suitable for monitoring therapeutic responses and detecting MRD [24]. For colorectal cancer, a screening tool based on hypermethylation detection through ctDNA is available. It shows acceptable sensitivity and specificity and is currently approved by the FDA for colorectal screening [25]. Moreover, the clinical relevance of ctDNA as a prognostic marker for the detection of MRD and recurrence after surgery has been shown [26]. For breast cancer, there are two FDA-approved prognostic tests using ctDNA liquid biopsy; ctDNA has also been explored for treatment response prediction and MRD detection [27,28,29].

ctDNA has been explored in hematological malignancies such as leukemias, multiple myeloma, and myelodysplastic syndromes, as well as in lymphomas [5,30]. Specific examples will be shown in the results portion of this paper.

### 1.3. Methods for ctDNA Detection in Peripheral Blood

ctDNA analysis in peripheral blood involves its extraction and subsequent detection through different techniques. Plasma is more suitable than serum for ctDNA isolation, and prompt centrifugation and minimal storage time are preferable to minimize genomic DNA (gDNA) contamination and cfDNA concentration decline, respectively [19,31]. cfDNA extraction is typically performed with commercial kits employing either the more established spin column-based approach or the more recently introduced magnetic bead-based approach [32].

After extraction, ctDNA can be analyzed through targeted or non-targeted methods, which are mainly based on PCR (polymerase chain reaction) and NGS (next-generation sequencing), respectively. Targeted approaches are useful for detecting and monitoring specific mutations, while non-targeted approaches may be used for the discovery of novel genes for precision medicine [19,33]. PCR-based technologies used for ctDNA detection include real-time PCR (qPCR), BEAMing (beads, emulsions, amplification, and magnetics), and digital droplet PCR (ddPCR), whereas NGS-based techniques include tagged amplicon deep sequencing (Tam-Seq), safe-sequencing system (Safe-SeqS), cancer personalized profiling by deep sequencing (CAPP-Seq), whole-genome sequencing (WGS), and whole-exome sequencing (WES) [33,34].

The low ctDNA blood concentration, especially in early-stage malignancies and in MRD, is the main limitation for its use as a biomarker. The need for extremely sensitive and time-consuming detection techniques makes its utilization in clinical practice often impractical and expensive [6,19]. To overcome this hurdle, many studies have proposed strategies to increase sensitivity and feasibility. Sonnenberg et al. developed a dielectrophoretic-based device for cfDNA isolation, which showed the same sequencing results as cfDNA extracted with a commercial kit [35]. Soscia et al. developed and validated a laboratory protocol to minimize pre-analytical interference in cfDNA analysis and evaluated a cfDNA amplification system with satisfactory results [36]. Beagan et al. analyzed a PCR-free shallow whole genome sequencing (sWGS) method that was comparable to traditional PCR methods. This result shows that a PCR-free sWGS method would be feasible in an oncological context with the advantage of a faster workflow [37]. Sampathi et al. developed a simple and rapid workflow for cfDNA monitoring based on Nanopore Minion sequencing of PCR-amplified B-cell-specific rearrangement of the immunoglobulin-heavy (IGH) locus in cfDNA from ALL patients. Its low cost and ease of use make it an attractive assay for clinical settings [38]. Wang et al. developed and validated a strategy to lower detection errors by optimizing unique molecular identifiers-based error suppression. This method could be used to boost mutation detection accuracy, and thus, improve the cost-effectiveness and clinical impact of NGS [39].

Circulating miRNA detection follows much the same logic as cfDNA analysis. Sample collection is preferably performed with EDTA or citrate as the anticoagulant, and extraction is performed by phenol/guanidinium or column-based methods [40]. miRNA may be analyzed through PCR-based techniques such as qPCR and TaqMan Array Cards or NGS, Northern Blotting, and microarray analysis [41,42]. Some studies have proposed new methods for detecting circulating miRNA in leukemia, improving sensitivity and identifying post-transcriptional modifications. Zheng et al. proposed a new enzyme-free quadratic SERS (surface-enhanced Raman spectroscopy) signal amplification approach for circulating miRNA detection in human serum. This method was able to quantify miRNA-21 in CLL patients with results comparable to qPCR [43]. Similarly, Pero-Gascon et al. described an on-line solid-phase extraction–capillary electrophoresis–mass spectrometry (SPE-CE-MS) method that was also able to differentiate CLL and control samples through miR-21-5p detection [42].

Exosomes may be isolated from peripheral blood through different techniques, such as ultracentrifugation, precipitation, immunoaffinity capture, and size-based isolation [33]. Alternative techniques have been proposed for exosome isolation in leukemias; for instance, Li et al. developed a colorimetric biosensor for the ultrasensitive detection of leukemia-derived exosomes that successfully distinguished leukemia patients from healthy controls, showing potential use in diagnosis [44]. Yin et al. created a PEG-based method using label-free surface-enhanced Raman scattering to isolate extracellular vesicles, with 90.0% accuracy, 90.9% precision, and 83.3% sensitivity for an AML cell line [45]. Xue et al. developed a multivalent, long-single-stranded aptamer with repeated units for exosome isolation and suggested its use in ALL studies [46].

In Table 1, we summarize relevant studies regarding new techniques for cfDNA, miRNA, and exosome analysis optimization in blood.

## 2. Methodology

To conduct the search for this article, the following terms were used: (“liquid biopsy” or “cfDNA” or ctDNA) and leukemia, “circulating microRNA” and “leukemia”, and “exosomes” and “leukemia” and “liquid biopsy”. The search was limited to the PubMed and ClinicalTrials.org databases, published from 2010 up to March 2025, in the English language. The original search retrieved 111 articles. Repeated articles and work concerning microbial cfDNA, blast cells, or pathologies different than leukemia (such as multiple myeloma and lymphoma) were excluded. Withdrawn or inapplicable clinical studies were also excluded. Using these criteria, fifty-two manuscripts (Figure 1) and fourteen clinical trials were subjected to an evaluation process.

## 3. Lymphoid Leukemias

### 3.1. Acute Lymphoblastic Leukemia (ALL)

Most studies regarding the use of liquid biopsy in ALL focused on miRNAs as biomarkers. Swellam et al. investigated the expression of miRNA-125b-1 and miRNA-203 in childhood ALL and considered them useful biomarkers for diagnosis [47]. Also, for diagnosis, Shahid et al. showed that miR-146a may be useful biomarker for both adults and children with ALL [48]. Luna-Aguirre et al. performed a miRNA expression profiling assay and found seventy-seven circulating miRNAs differentially expressed between ALL samples and controls. MiR-511 was the most valuable biomarker for distinguishing B-ALL from normal controls [49].

Regarding cfDNA detection, a pilot study conducted on a cohort of twenty-five patients diagnosed with AML and ALL demonstrated that the evaluation of cfDNA exhibited a higher degree of accuracy in predicting poor outcomes in comparison with the conventional method of MRD detection by flow cytometry [50]. In a case report, Luskin et al. used cfDNA sequencing to detect fetal aneuploidy, revealing a chromosomal anomaly that was later correlated with a case of maternal ALL [51].

Liquid biopsy can be used as a tool for treatment choice, monitoring of effectiveness, and response prediction in ALL. Kobayashi et al. developed a liquid biopsy method based on whole-blood imaging flow cytometry to evaluate the susceptibility of leukemic cell lines to different drugs, hoping to use such a tool in the precision medicine field [52]. In a case report by Medinger et al., liquid biopsy was used to monitor treatment response to a novel compound in a patient with relapsed and refractory ALL. ctDNA levels and the presence of relevant gene variants were correlated with treatment response [53].

Liquid biopsy may also be used as a tool to detect minimal residual disease (MRD) in ALL patients. The studies regarding this topic explored ctDNA and miRNA as biomarkers for MRD. Aljurf et al. proposed the use of ctDNA analysis for early relapse prediction in patients post-hematopoietic stem cell transplant (HSCT). ctDNA was more sensitive to early relapse prediction than classic cellular DNA analysis [54]. A proof-of-concept study by Arthur et al. also utilized ctDNA to monitor MRD in ALL patients, using patient-specific targets determined from whole-genome sequencing [55]. Somatic variants (sequence mutations, copy number variations, and structural variations that result in oncogenic fusions) from ctDNA were correlated with MRD by Lei et al. Analysis of deep sequencing revealed heterogeneous ctDNA alterations in the plasma of children with hematologic malignancies [56]. Du et al. utilized next-generation sequencing (NGS) to observe the alignment of plasma ctDNA profiles with the mutational profile of bone marrow samples and MRD in pediatric acute lymphocytic leukemia (ALL), thereby underscoring the potential of ctDNA as a noninvasive instrument for disease monitoring and predicting therapy in B-ALL [57]. miRNAs were also explored as biomarkers for MRD in childhood ALL, such as miR-128-3p and miR-222-3p [58,59].

Liquid biopsy can also be useful for monitoring toxicity and specific disease complications such as central nervous system involvement [60,61]. For instance, Totoń-Żurańska et al. correlated miRNA markers with doxorubicin cardiotoxicity [62]. Kumar et al. hypothesized that cfDNA levels contribute to heightened thrombin generation potential in ALL patients, leading to thromboembolism [63]. The studies in ALL using liquid biopsy are summarized in Table 2.

### 3.2. Chronic Lymphoid Leukemia (CLL)

Eight studies using liquid biopsy in CLL were assessed. A study by Lenaerts et al. performed genome-wide screening of cfDNA in healthy elderly people and found altered copy number alterations (CNAs) in a fraction (3%) of the population. A six-month follow-up and magnetic resonance imaging (MRI) exam confirmed malignant diagnosis in five patients, including a CLL case [64]. A nested case-control study within the European Prospective Investigation into Cancer and Nutrition (EPIC) examined the potential of five miRNAs as predictive of CLL. MiR-29a, miR-150-5p, and miR-155-5p were upregulated in early stages of CLL but had limited ability to discriminate between pre-CLL and controls [65]. Additionally, Hemenway et al. reported a prostate cancer patient who, through the analysis of ctDNA mutations, was later diagnosed with additional hairy-cell leukemia [66]. Another case reported a pregnant woman who was diagnosed with CLL through non-invasive prenatal testing, which analyzes cfDNA circulating in maternal blood [67].

Regarding treatment monitoring, Yeh et al. found that ctDNA reflects changes in disease burden across different tissue compartments in response to treatment in CLL patients using ibrutinib or venetoclax [68]. Albitar et al. made use of liquid biopsy to detect mutations related to treatment resistance and found plasma cfDNA more reliable than cellular DNA in detecting such mutations [69]. Kurtz et al. applied liquid biopsy in the context of outcome prediction by developing a risk index method (Continuous Individualized Risk Index, CIRI) which uses three ctDNA risk factors as part of its model. When CIRI was applied in CLL patients, it was able to satisfactorily stratify patients regarding outcomes and overall survival [70].

Stamatopoulos et al. compared miRNA-150 expression in the serum and CD19+ cells of CLL patients and found that a low cellular miR-150 expression level was associated with tumor burden, disease aggressiveness, and poor prognostic factors, whereas a high level of serum miR-150 was associated with tumor burden markers and some markers of poor prognosis. Due to these results, the authors suggest that circulating serum miRNA-150 could be a biomarker for disease progression and patient stratification [71]. Studies on CLL employing liquid biopsy techniques are summarized in Table 3.

## 4. Myeloid Leukemias

### 4.1. Acute Myeloid Leukemia (AML)

“Methylated” (5-hydroxymethylcytosine) cfDNA and exosomal long non-coding RNAs have been proposed as biomarkers for the diagnosis and monitoring of AML [72,73]. Ruan et al. concluded through a comparison study that ctDNA mirrors the genomic information from the bone marrow, having potential application as a diagnostic biomarker [74]. Shao et al. suggest that 5-hydroxymethylcytosine cfDNA could also be used as a tool to stratify AML patients in risk groups [75]. In the context of acute promyelocytic leukemia, PML-RARA as found in extracellular vesicles has been explored as a potential diagnosis tool [76].

Some studies have utilized liquid biopsy in the diagnosis of AML as a secondary condition. For instance, a liquid biopsy uncovered a case of AML in a lung cancer patient through specific mutations in cfDNA [77]. It was also able to detect the onset of therapy-related acute leukemia in a Diffuse Large B Cell Lymphoma patient [78].

In regard to the application of liquid biopsy for treatment monitoring, Koutova et al. found six miRNAs that were responsive to chemotherapy in AML patients [79]. Zeidan et al. found that ctDNA mirrored the mutant allele frequency found in the bone marrow, and its levels were predictive of treatment response [80]. Similarly, a study by Zhou et al. concluded that ctDNA can reflect treatment response, survival, and clonal evolution in AML [81]. Yadav et al. proposed the use of liquid biopsy to monitor and predict the response to total body irradiation (TBI) in acute leukemia patients. The authors found that miR150 acted as a marker for treatment response in mice and patient cells [82].

Minimal residual disease has been detected and monitored with liquid biopsy in AML patients. Gao et al. showed that the concentration and integrity of ctDNA reflects complete remission status and minimal residual disease in both AML and ALL patients, being a potential biomarker for MRD monitoring [83]. Zhong et al. explored the application of liquid biopsy in monitoring monoclonal immunoglobulin heavy chain (IGH) and T-cell receptor (TCR) rearrangements through ctDNA, which can be used as markers for MRD [84]. Short et al. also used cfDNA as a possible MRD marker in a study comparing the mutations found in cfDNA and bone marrow [85]. Rausch et al. aimed to improve the sensibility and practicality of liquid biopsy by developing a double drop-off droplet digital PCR assay for specific gene mutations in AML [86]. Hourigan et al. made use of liquid biopsy to detect MRD in a study regarding post-allogenic hematopoietic cell transplant treatment. They concluded that patients with detectable MRD would be better served by myeloablative conditioning to prevent relapses [87]. Mata et al. analyzed liquid biopsy-based NGS of ctDNA and identified characteristic genetic alterations in hematopoietic neoplasms, including FLT3, IDH2, and NPM1, in AML patients, and across the study, 75% of samples exhibited alterations exclusively in liquid biopsy compared with paired buffy coat, bone marrow, or tissue samples [88].

Liquid biopsy has been explored as a prognostic prediction tool for AML patients. In a study by Yegin et al., lower levels of ctDNA in a pre-transplant stage correlated with post-transplant relapse [89]. Nakamura et al. found ctDNA a reliable marker for relapses in patients post-allogenic stem cell transplant [90]. Regarding miRNA expression, Tian et al. found that low miRNA-192 in serum was correlated with aggressive clinical features and could be used as an independent prognostic indicator for overall survival and event-free survival [91]. Similarly, low expression of circulating miRNA-328 was also associated with poor prognosis in a study by Liu et al. [92].

Some studies used liquid biopsy to predict leukemia complications. For instance, Purhonen et al. [93] found that the cfDNA/leukocyte ratio was able to predict the development of neutropenic fever with complicated courses in AML patients.

Liquid biopsy may be useful as a tool in mutation studies. Anyanwu et al. analyzed cfDNA for the presence of NRAS gene mutations in codons 12 and 13 in hematological malignant patients, including AML, and healthy controls. cfDNA was chosen over peripheral blood mononuclear cells (PBMCs) or blood marrow due to its more extensive mutational detection [94]. In Table 4, we summarize the studies found that used liquid biopsy in AML.

### 4.2. Chronic Myeloid Leukemia (CML)

The literature regarding liquid biopsy application in CML was scarce. A novel dPCR platform was validated with BCR-ABL1 transcripts to prove its potential use in precision medicine [95]. Additionally, Ferreira et al. compared the expression of 768 miRNAs in peripheral blood in newly diagnosed and imatinib-treated chronic-phase CML patients and found eighty deregulated miRNAs involved in different pathways [96]. Similarly, Martins et al. compared the expression profile of circulating miRNAs in newly diagnosed and post-allogenic hematopoietic stem cell transplant CML patients and identified forty-six differentially expressed miRNAs [97]. Bernardi et al. explored the feasibility of tumor-derived exosome enrichment in CML through the quantification of the BCR-ABL1 exosomal transcript by digital PCR. In the future, such a method could be applied to MRD detection [98].

## 5. Clinical Trials

Currently, there are 14 clinical trials involving liquid biopsy in leukemias registered on ClinicalTrails.gov. Seven of these are large-scale trials regarding the use of precision medicine in cancer [99,100,101,102,103,104,105]. Four studies look to correlate ctDNA mutation status in response to novel treatments [106,107,108,109]. One study aims to assess the utility of cfDNA as a biomarker for the early detection of treatment resistance [110]. Another study evaluating the integration of ctDNA-based analyses into flow cytometry for MRD assessment should improve the early detection of relapses [111]. Finally, a trial in Beijing will evaluate the clinical utility of ctDNA for acute leukemia patients after chemotherapy [112]. Relevant clinical trials regarding leukemias are summarized in Table 5.

## 6. Discussion

We identified several studies on using liquid biopsy for diagnosis, monitoring, and minimal residual disease (MRD) detection in acute leukemias, and to a lesser extent, in chronic lymphoid leukemia (CLL). Among these, MRD detection was the most frequently explored application across the leukemia types, followed by treatment monitoring.

For acute lymphoblastic leukemia (ALL), MRD monitoring plays a critical role in tailoring therapy protocols and assessing treatment responses [113,114]. In adult ALL, which is less common but is associated with poorer outcomes, relapse occurs in up to 50% of cases [109]. Thus, MRD serves as a crucial prognostic indicator in both pediatric and adult ALL. More than half of the studies on liquid biopsy in ALL focused on MRD, reflecting the demand for highly sensitive and accurate methods to detect and monitor residual disease. Studies from Aljurf et al., Arthur et al., and Rzepiel et al. compared liquid biopsy-based MRD monitoring with conventional bone marrow and circulating cell analyses, finding comparable sensitivity and accuracy, thereby supporting its potential clinical application [54,55,58,59].

Acute myeloid leukemia (AML) encompasses a group of heterogeneous malignancies with diverse clinical and molecular characteristics. Consequently, treatment sensitivity varies and is influenced by both patient-specific factors and disease biology [115]. This variability contributes to frequent clinical relapses, underscoring the need for robust tools to monitor treatment response and MRD. Studies have explored the utility of exosomal long non-coding RNAs (lncRNAs), circulating tumor DNA (ctDNA), and microRNAs (miRNAs) as biomarkers for treatment monitoring, with ctDNA also being investigated for MRD assessment [73,79,80,82,83,84,85,86,87]. These findings suggest the potential of liquid biopsy to address the clinical challenges posed by AML’s heterogeneity.

While MRD analysis is well established in ALL for guiding therapeutic decisions, its application in chronic lymphocytic leukemia (CLL) has only recently been introduced [116]. This review uncovered a limited number of studies investigating liquid biopsy for MRD detection in CLL, presenting a promising area for future research.

For chronic myeloid leukemia (CML), unlike other leukemias, there is a notable lack of studies evaluating the application of liquid biopsy, highlighting a gap in current research. This may be because follow-up protocols to detect levels of the bcr-abl are well established throughout the world.

The integration of precision medicine into both solid and hematological malignancies has received significant attention in recent years [117,118,119,120,121]. The clinical trials and studies explored in this review reflect this trend, particularly in leukemias. Thirteen clinical trials involving liquid biopsy were identified; they are being conducted on CLL and AML patients, with some including patients with any leukemia type. A growing interest in personalized treatment for childhood cancers and the use of prognostic biomarkers to optimize therapy is evident.

ctDNA emerged as the most studied biomarker in both published studies and clinical trials, consistent with its advantages over other biomarkers. These advantages include highly sensitive assays, a short half-life that enables real-time monitoring of the tumor landscape, and its non-invasive nature [7,8]. miRNAs were also frequently investigated, given their stability and significant role in leukemogenesis [79,82]. Interestingly, ongoing clinical trials in CLL predominantly focus on ctDNA monitoring in novel therapies to assess efficacy and safety, a topic less emphasized in the existing literature. However, as these trials are still ongoing, their concordance with published findings remains uncertain.

Most studies included herein had small cohort sizes, with over half of them involving fewer than 50 patients and only a few exceeding 200 participants. Consequently, the promising results reported would benefit from validation in larger, multicenter studies. The ongoing clinical trials will be pivotal in determining the clinical utility of liquid biopsy in leukemias. Technical challenges, such as insufficient sensitivity of the assays and high costs, were frequently cited as barriers to broader implementation. Efforts to develop cost-effective workflows and reduce error rates, as highlighted in some studies [38,39], represent critical avenues for future research.

This review has limitations, including its reliance only on studies published in English and indexed in PubMed, which may have excluded relevant non-English or inaccessible research. Despite these constraints, our findings suggest that liquid biopsy holds great promise for MRD detection and treatment monitoring in leukemias. Continued technological advancements and larger-scale studies will likely address the current hurdles, facilitating its clinical adoption.

## 7. Conclusions

Liquid biopsy methods have demonstrated significant potential for evaluating treatment response and MRD monitoring in leukemia. As techniques evolve, they have the potential to become widely adopted in clinical settings, either as standalone tools or in combination with other currently used methods for the management of patients with leukemia and other hematological diseases. However, before a complete integration of liquid biopsy into clinical practice, key challenges remain, including standardizing cfDNA extraction protocols, enhancing the sensitivity of NGS- and PCR-based assays, and achieving widespread validation through clinical trials. Addressing these challenges will be essential for the successful integration of liquid biopsy into routine hematological clinical practice.

## Figures and Tables

**Figure 1 cancers-17-01438-f001:**
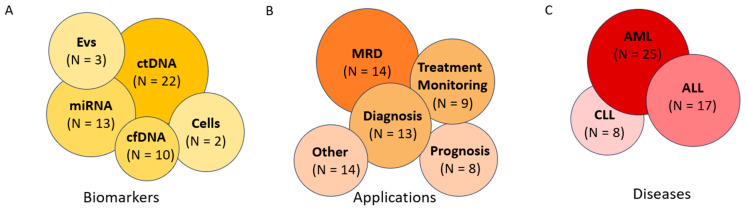
Summary of current studies exploring liquid biopsy in leukemias investigated in this review. (**A**) Biomarkers, (**B**) applications, and (**C**) diseases in studies utilizing liquid biopsy in leukemia. EVs: extracellular vesicles; ctDNA: circulating tumor DNA; cfDNA: cell-free DNA, miRNA: micro-RNA; MRD: minimal residual disease; ALL: acute lymphoid leukemia; AML: acute myeloid leukemia; CLL: chronic lymphoid leukemia.

**Table 1 cancers-17-01438-t001:** Description of relevant studies regarding analysis optimization in leukemias.

Reference	Disease	Key Points
Beagan et al., 2021 [37]	AML	PCR-free sWGS can satisfactorily detect somatic mutations in cfDNA.
Li et al., 2022 [44]	AML	Colorimetric biosensor can distinguish healthy and leukemia patients through exosome detection.
Pero-Gascon et al., 2018 [42]	CLL	SPE-CE-MS method was able to differentiate CLL patients and control through microRNA detection.
Sampathi et al., 2022 [38]	ALL	Nanopore IGH detection workflow is a simple and inexpensive assay for cfDNA monitoring in ALL.
Sonnenberg et al., 2014 [35]	CLL	cfDNA was rapidly isolated from peripheral blood through a DEP device.
Soscia et al., 2022 [36]	ALL, AML, CML, CLL	An amplification system was proved to be free of analytical biases and efficient at increasing ctDNA amounts at diagnosis and in follow-up samples.
Wang et al., 2019 [39]	AML	The Singleton Correction method can be incorporated into UMI-based error suppression workflows to increase accuracy.
Xue et al., 2021 [46]	ALL	Biotin-labeled multivalent aptamers can be used to efficiently and specifically isolate EVs derived from malignant lymphocytes.
Yin et al., 2021 [45]	AML	PEG-based isolation of extracellular vesicles is a simple and low-cost assay for exosome detection in peripheral blood.
Zheng et al., 2015 [43]	CLL	An enzyme-free quadratic signal amplification strategy via target miRNA-triggered hybridization chain reaction and ions-mediated cascade amplification is able to detect circulating miRNA in CLL patient serum.

ALL, acute lymphoid leukemia; AML, acute myeloid leukemia; cfDNA, cell-free DNA; CLL, chronic lymphoid leukemia; CML, chronic myeloid leukemia; DEP, dielectrophoresis; miRNA, microRNA; SPE-CE-MS, solid-phase extraction–capillary electrophoresis–mass spectrometry; sWGS, shallow whole-genome sequencing; UMI, unique molecular identifier; IGH, immunoglobulin heavy chain.

**Table 2 cancers-17-01438-t002:** Description of relevant studies applying liquid biopsy in ALL.

Reference	Diseases	Analyte	N	Method	Key Points	Application
Aljurf et al., 2016 [54]	ALL, AML, and CML	ctDNA	126	STR-PCR	cfDNA-based chimerism testing in patients after HSCT might be more reliable than standard cell subset testing method.	Minimal residual disease
Arthur et al., 2022 [55]	ALL	ctDNA	6	WGS and ddPCR	cfDNA extracted from blood and cerebrospinal fluid was used to monitor MRD with sensitivity comparable to that in bone marrow.	Minimal residual disease
Du et al., 2024 [57]	ALL	ctDNA	146	NGS	cfDNA from peripheral blood and bone marrow. MRD measured on day 19. Novel fusions found. Elevated mutation counts and maximum variant allele frequency in baseline BM were associated with significantly poorer chemotherapy response.	Minimal residual disease
Egyed et al., 2022 [60]; Egyed et al., 2020 [61]	ALL	miRNA	9	ELISA	VEGF-A and miR-181a expression were increased in CNS compromised patients; joint quantification of miR-181a and VEGF-A might provide a novel tool to precisely diagnose CNS involvement.	CNS involvement
George et al., 2024 [50]	ALL/AML	cfDNA	25	cfDNA levels (ratio baseline: follow-up)	cfDNA ratio score of 2.6 or higher at diagnosis/remission predicted poor outcomes with higher accuracy than conventional MRD detection by flow cytometry.	Outcome, treatment response
Kobayashi et al., 2019 [52]	ALL	CTCs	3	Whole-blood imaging flow cytometry	The developed method was able to accurately evaluate drug susceptibility via dose-dependent morphological changes in WBCs.	Treatment selection
Kumar et al., 2021 [63]	ALL	cfDNA	17	Fluorometric DNA quantitation	cfDNA contributes to increased thrombogenic potential in ALL patients.	Thromboembolism
Lei et al., 2025 [56]	Childhood leukemias	ctDNA	177	Pen-Seq (deep sequencing) for somatic genomic variants	Somatic variants (sequence mutations, copy number variations, and structural variations responsible for oncogenic fusions) from ctDNA correlated with minimal residual disease.	Diagnosis and minimal residual disease
Luna-Aguirre et al., 2015 [49]	ALL	miRNA	39	qPCR	hsa-miR-511, -222, and -34a were overexpressed, whereas hsa-miR-199a-3p, -223, -221, and -26a were underexpressed in ALL samples.	Diagnosis
Luskin et al., 2017 [51]	ALL	cfDNA	1	Massively Parallel Shotgun Sequencing	Normalized DNA copy number estimations on chromosome 21 from cfDNA derived from maternal plasma demonstrated an amplification that was later correlated with iAMP21 B-cell ALL.	Diagnosis
Medinger et al., 2022 [53]	ALL	ctDNA	1	NGS	NGS of the ctDNA indicated residual disease before allo-HSCT, and non-detectable gene variants suggested disease control after allo-HSCT.	Treatment monitoring
Rzepiel et al., 2019 [58]	ALL	miRNA	20	qPCR	MiR-128-3p and miR-222-3p in blood predict day 15 flow cytometry MRD results 7 days earlier.	Minimal residual disease
Rzepiel et al., 2023 [59]	ALL	miRNA	13	qPCR	miR-128-3p expression positively correlates with bone marrow MRD on the 15th day of treatment, so it might be a valuable biomarker for following the bone marrow function or the therapy response in ALL.	Minimal residual disease
Shahid et al., 2021 [48]	ALL	miRNA	66	qPCR	Circulating miR-146a is upregulated in ALL, and its expression level significantly decreased after treatment.	Diagnosis and treatment monitoring
Swellam et al., 2018 [47]	ALL	miRNA	43	qPCR	miRNA-125b-1 is overexpressed, and miRNA-203 is downregulated in childhood ALL with high diagnostic value.	Diagnosis
Totoń-Żurańska et al., 2022 [62]	ALL	Exosomal miRNAs	66	NGS	miRNA levels reflect doxorubicin-induced myocardial injury and preceded development of late-onset cardiomyopathy phenotype.	Prognosis

ALL, acute lymphoid leukemia; allo-HSCT, allogeneic hematopoietic stem cell transplant; AML, acute myeloid leukemia; cfDNA, circulating cell-free DNA; CML, chronic myeloid leukemia; CNS, central nervous system; CTCs, circulating tumor cells; ctDNA, circulating tumor DNA; ddPCR, digital droplet PCR; ELISA, enzyme-linked immunosorbent assay; HSCT, hematopoietic stem cell transplant; miRNA, microRNA; MRD, minimal residual disease; NGS, next-generation sequencing; qPCR, quantitative PCR; STR-PCR, short tandem repeat–polymerase chain reaction; WBC, white blood cell; WGS, whole-genome sequencing.

**Table 3 cancers-17-01438-t003:** Description of relevant studies applying liquid biopsy in CLL.

Reference	Disease	Analyte	N	Method	Key Points	**Application**
Albitar et al., 2017 [69]	CLL	ctDNA	16	HS assay with WTB-PCR, Sanger sequencing and NGS	HS assay detected resistance-related mutations; plasma cfDNA more sensitive than cellular DNA or serum cfDNA.	Treatment monitoring
Casabonne et al., 2020 [65]	CLL	miRNA	224	qPCR	Circulating hsa-miRNA-29a-3p, hsa-miR-150-5p, and hsa-miR-155-5p deregulated up to 10 years before CLL diagnosis.	Evaluating the role of miRNAs in CLL pathogenesis
Di Giosaffatte et al., 2022 [67]	CLL	cfDNA	1	NIPT	NIPT result and hemogram contributed to CLL diagnosis of pregnant woman.	Diagnosis
Hemenway et al., 2022 [66]	HCL	ctDNA	1	NGS	RAF V600E and CHEK2 mutations identified by liquid biopsy had a key role in HCL diagnosis.	Diagnosis
Kurtz et al., 2019 [70]	CLL	ctDNA	1426	Naïve Bayes	A risk index was developed and validated to improve accuracy of prognostic models and facilitate treatment choice.	Prognosis
Lenaerts et al., 2019 [64]	CLL	ctDNA	1002 healthy individuals	WGS and GIPseq	Out of the 1002 screened participants, 5 were diagnosed with hematological cancer after cfDNA CNA analysis.	Screening
Stamatopoulos et al., 2015 [71]	CLL	miRNA	273/252	qPCR	Cellular and serum levels of miR-150 are associated with opposite clinical prognoses and could be used to molecularly monitor disease evolution in CLL.	Prognosis
Yeh et al., 2017 [68]	CLL	ctDNA	32	TS and dPCR	ctDNA dynamics reflect changes in disease burden across different tissue compartments.	Treatment monitoring

CLL, chronic lymphoid leukemia; CNA, copy number alteration; ctDNA, circulating tumor DNA; dPCR, digital PCR; GIPseq, genomic imbalance profiling from cfDNA sequencing; HCL, hairy-cell leukemia; HS, high sensitivity; miRNA, microRNA; NGS, next-generation sequencing; NIPT, non-invasive prenatal test; qPCR, quantitative PCR; TS, targeted sequencing; WGS, whole-genome sequencing; WTB-PCR, wild-type blocking polymerase chain reaction.

**Table 4 cancers-17-01438-t004:** Description of relevant studies applying liquid biopsy in AML.

Reference	Disease	Analyte	N	Method	Key Points	Application
Anyanwu et al., 2019 [94]	Multiple	cfDNA	200	Multiplex allele-specific PCR	cfDNA was used to detect NRAS G12D and NRAS G13C mutations in Federal Capital Territory, Nigeria.	Mutation studies
Barzegar et al., 2021 [76]	APL	Extracellular vesicles	22	RT-PCR, qPCR, and flow cytometry	Plasma-derived EVs contain PML-RARα fusion transcript.	Diagnosis
Gao et al., 2010 [83]	ALL and AML	ctDNA	44 (AML) 16 (ALL)	qPCR	DNA concentrations and integrity were significantly higher in cancer patients, and DNA integrity at CR had a distinct reduction and then an increase at relapse.	Minimal residual disease
George et al., 2024 [50]	ALL/AML	cfDNA	25	cfDNA levels (ratio baseline/follow-up)	cfDNA ratio score of 2.6 or higher at diagnosis/remission predicted poor outcomes with higher accuracy than conventional MRD detection by flow cytometry.	Outcome, treatment response
Hourigan et al., 2020 [87]	AML	ctDNA	190	NGS	Liquid biopsy was used to evaluate MRD.	Minimal residual disease
Kerle et al., 2022 [78]	AML	ctDNA	1	Targeted NGS and ddPCR	NGS and ddPCR combination was able to detect AML onset through specific mutations.	Diagnosis
Khoo et al., 2019 [95]	AML	Blast cells	15	BCB	Enrichment of blast cells was achieved from blood with a one-step microfluidic blast cell biochip (BCB) sorting system.	Minimal residual disease
Koutova et al., 2015 [79]	AML	miRNA	8	qPCR	miR-199b-5p, miR-301b, miR-326, miR-361-5p, miR-625, and miR-655 levels were sensitive to therapy.	Treatment monitoring
Liu et al., 2015 [92]	AML	miRNA	176	qPCR	Serum miRNA-328 was significantly downregulated in AML patients compared with healthy controls and was an independent prognostic factor for OS and RFS.	Prognosis
Mata et al., 2024 [88]	AML	ctDNA	10	NGS	Alterations found in FLT3, IDH2, and NPM1. Across the study, 75% of samples showed alterations only in liquid biopsy (in comparison with paired buffy coats, marrow, or tissues).	Minimal residual disease, existence of treatment-resistant clones
Nakamura et al., 2019 [90]	AML	ctDNA	37	NGS and ddPCR	Patients with persistent ctDNA+ status at 1 or 3 post-alloSCT had a significantly higher risk of relapse; increasing ctDNA levels between 1 month and 3 months post-alloSCT was the precise predictor of relapse.	Prognosis
Nkosi et al., 2022 [77]	AML	ctDNA	1	NGS	Lung cancer patient was diagnosed with AML after liquid biopsy results showed suggestive mutations.	Diagnosis
Purhonen et al., 2015 [93]	AML	cfDNA	32	qPCR	cfDNA/leukocyte ratio on d0 predicted complicated course of neutropenic fever (*p* = 0.019) with area under the curve (AUC) 0.76 and 95% confidence interval (CI 0.54–0.98).	Complications
Rausch et al., 2021 [86]	AML	ctDNA	12	DDO-ddPCR	Novel ddPCR assay design was able to detect AML mutations with sensitivity equal to qPCR.	Minimal residual disease
Ruan et al., 2021 [74]	AML	cfDNA	20	NGS	Consistency analysis showed that ctDNA can mirror the genomic information from the bone marrow, and a subset of mutations was exclusively detected in ctDNA.	Diagnosis
Shao et al., 2022 [72]	AML	ctDNA	103	nano-hmC-Seal-Seq	Plasma cfDNA 5hmC levels change dynamically with disease burden in AML.	Diagnosis and prognosis
Shao et al., 2023 [75]	AML	cfDNA	54	NGS	5hmC levels in the regions of H3K4me3 in cfDNA were used to group AML patients in 3 groups with particular characteristics.	Patient stratification
Short et al., 2020 [85]	AML	ctDNA	22	NGS	Sequencing cfDNA can identify clinically relevant mutations not detected in the bone marrow.	Minimal residual disease
Tian et al., 2018 [91]	AML	miRNA	97	qPCR	Serum miRNA-192 could be a reliable biomarker for AML diagnosis and prognosis.	Prognosis
Xiao et al., 2022 [73]	AML	Exosomal lncRNAs	65	qPCR	LINC00265, LINC00467, UCA1, and SNHG1 were able to distinguish AML patients from control, and their combined use exhibited the most powerful diagnostic accuracy.	Diagnosis and treatment monitoring
Yadav et al., 2022 [82]	AML	miRNA	22	qPCR	miR150 levels were decreased after radiation and exhibited an inverse correlation with recurrence.	Treatment monitoring
Yegin et al., 2020 [89]	AML and ALL	ctDNA	99 (AML) 74 (ALL)	Spectrophotometry	Lower levels of pre-transplant cfDNA seem to be associated with transplant-related morbidities.	Prognosis
Zeidan et al., 2020 [80]	AML	ctDNA	40	NGS and ddPCR	Decrease in mutant ctDNA during therapy was associated with clinical response.	Treatment monitoring
Zhong et al., 2018 [84]	AML	ctDNA	235	qPCR	Monoclonal IGH and TCR rearrangement were detected in cfDNA.	Minimal residual disease
Zhou et al., 2023 [81]	AML	ctDNA	14	NGS	Serial plasma-derived ctDNA assessments can reflect treatment response, survival, and clonal evolution in AML.	Treatment monitoring

5hmC, 5-Hydroxymethylcytosine; ALL, acute lymphoid leukemia; alloSCT, allogeneic stem cell transplantation; AML, acute myeloid leukemia; ctDNA, circulating tumor DNA; DDO-ddPCR, double drop-off droplet digital PCR; ddPCR, digital droplet PCR; lncRNA, long non-coding RNA; miRNA, microRNA; NGS, next-generation sequencing; OS, overall survival; qPCR, quantitative PCR; RFS, relapse-free survival; RT-PCR, reverse transcription PCR.

**Table 5 cancers-17-01438-t005:** Description of relevant clinical trials involving liquid biopsy in leukemias.

ClinicalTrials.gov Identifier	Disease	N	Recruitment Status	Brief Summary	Liquid Biopsy Application
NCT03892044, 2019 [107]	CLL	7	Active, not recruiting	Will determine the maximum tolerated dose (MTD) of duvelisib in combination with nivolumab for patients with Richter’s syndrome or transformed follicular lymphoma.	Response to duvelisib in combination with nivolumab will be correlated with DNA mutation of CLL and lymphoma and assessed in tumor samples and cell-free DNA.
NCT05504772, 2022 [102]	AML and ALL	3500	Recruiting	Aims to improve outcomes for childhood cancer patients through the implementation of precision medicine.	Liquid biopsy will be investigated as a non-invasive method for diagnosis of tumors that are difficult to biopsy directly.
NCT05168904, 2021 [108]	AML and CLL	210	Recruiting	Two-part, phase 1/2, open-label, multicenter study designed to evaluate the safety and efficacy of fadraciclib administered orally BID.	To investigate plasma cell-free DNA mutation and copy number variation profile of fadraciclib as determined by NGS.
NCT03702309, 2018 [103]	Leukemia	2500	Recruiting	Means to develop an institution-wide liquid biopsy protocol for future research studies at the University Health Network’s Princess Margaret Cancer Centre.	Peripheral blood samples will be incorporated into research protocols.
NCT03496402, 2018 [100]	AML	600	Recruiting	Means to identify and characterize meaningful molecular genetic alterations and immunological features of high-risk childhood, adolescent, and young adult cancers at diagnosis, during patient treatment, and at follow-up.	Molecular biology techniques will be used to identify potential prognostic biomarkers on samples collected during patients’ treatment and follow-up based on changes in ctDNA.
NCT03336931, 2017 [101]	AML and ALL	550	Recruiting	Multicenter prospective study of the feasibility and clinical value of a diagnostic service for identifying therapeutic targets and recommending personalized treatment for children and adolescents with high-risk cancer.	Laboratory analysis will include liquid biopsy.
NCT04790045, 2021 [109]	CLL	30	Not yet recruiting	Will evaluate lymph node changes in CLL patients treated with venetoclax-based regimens at molecular and ultrasound levels.	Peripheral blood samples will be drawn for pharmacodynamics studies, MRD assessment, and cfDNA evaluation.
NCT05366881, 2022 [104]	Leukemia	7000	Recruiting	Observational case–control study to train and validate a genome-wide methylome enrichment platform to detect multiple cancer types and to differentiate among cancer types.	cfDNA analysis will be used to differentiate cancer signals from specific cancer types and identify the correct tissue of origin.
NCT04898894, 2021 [110]	AML	42	Recruiting	Will determine the safety and tolerability of selinexor and venetoclax in combination with chemotherapy in pediatric patients with relapsed or refractory AML or ALAL.	Will explore associations between leukemia cell genomics, BCL2 family member protein quantification, BH3 profiling, and response to therapy as assessed by MRD and variant clearance using cfDNA.
NCT05691608, 2023 [105]	Leukemia	1800	Recruiting	Means to provide clinical therapeutic recommendations based on molecular profiling.	ctDNA analysis will be used to guide treatment strategy.
NCT02827617, 2016 [112]	CLL	56	Active, not recruiting	Means to identify molecular markers that can help the early and real-time prediction of response to ibrutinib treatment in CLL harboring TP53 mutations.	cfDNA for the identification of BTK and PLCγ2 resistance mutations will be compared with tumor genomic DNA genotyping.
NCT05099068, 2021 [106]	CLL	500	Recruiting	Prospective, multi-cohort study aiming to decipher molecular profiles/biological characteristics of advanced cancer patients during their disease with longitudinal and sequential analyses of tumor and liquid biopsies.	Means to identify potential prognostic and predictive biomarkers based on changes in ctDNA.
NCT04157569, 2019 [113]	AML	200	Recruiting	This study will monitor ctDNA through the ddPCR method in elderly acute leukemia patients after chemotherapy.	Will evaluate the clinical value of ctDNA.
NCT03787264, 2024 [111]	CLL	46	Active, not recruiting	This study evaluates the measurable residual disease (MRD)–guided triple combination of acalabrutinib, venetoclax, and obinutuzumab after optional bendamustine debulking in 45 patients with relapsed/refractory CLL.	The integration of ctDNA-based analyses into flow cytometry for MRD assessment is expected to enhance the early detection of relapses.

ALAL, acute leukemia of ambiguous lineage; ALL, acute lymphoid leukemia; AML, acute myeloid leukemia; BID, bis in die, twice a day; cfDNA, circulating cell-free DNA; CLL, chronic lymphoid leukemia; ctDNA, circulating tumor DNA; ddPCR, digital droplet PCR; MRD, minimal residual disease; NGS, next-generation sequencing.

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
