# Peer review of "Advancing Leukemia Management Through Liquid Biopsy: Insights into Biomarkers and Clinical Utility"

_cancers, 2025, doi:10.3390/cancers17091438_

Round 1

Reviewer 1 Report

Comments and Suggestions for Authors

The recent improvement in therapeutic tools able to enhance response rate, make urgent the identification of standardized methods for refining diagnosis and monitoring minimal residual disease.

In this paper the authors provided an accurate and updated review of the current use of liquid biopsy in management of acute and chronic leukemia.

The paper is clear, background and differences among biomarkers well described, as well as the limits of each method of evaluation.  Moreover, the paper also summarizes the current translational applications in myeloid and lymphoid diseases.

I have just an observation: It would be interesting provide an evaluation of the response time and of costs for the monitoring of each potential biomarker.

Author Response

We thank the reviewer for his valuable comments.

Comments 1: I have just an observation: It would be interesting provide an evaluation of the response time and of costs for the monitoring of each potential biomarker.

Response 1: This is a good suggestion. However, for the sake of clarity, the scope of this review does not include an evauation of costs.

Reviewer 2 Report

Comments and Suggestions for Authors

The authors review current use of liquid biopsy in management of AML, ALL and CLL. Under the label of "liquid biopsy" they mainly include measurements of cfDNA, ctDNA, and miRNA with only few instances of CTC or extracellular vesicles (exosomes) and no measurement of mRNA or metabolites. Therefore, the review largely discusses methodology and results of cfDNA, ctDNA and miRNA as components of potentially clinically useful liquid biopsy in leukemias. ctDNA detection and analysis and miRs are the main topics of this review. Some of the more recent technologies for analysis of cfDNA and miRs are indicated in Table 1, but it is not made clear how much better these methods are fin terms of their use as biomarkers.  Attempts are made to illustrate applications of liquid biopsy in leukemias to detection of MRD, diagnosis, prognosis and treatment monitoring. For AML, ALL and CTL tables are provided listing individual analytes, methods of detection, application to disease and short summary of results. The latter are descriptive, and while the relevant references are provided, the clinical significance of each study is not adequately covered. Most of the studies include small cohorts of patients and use different methods for each biomarker detection.  Thus, the clinical usefulness of each study, and the value of each biomarker as liquid biopsy in leukemias emerges as descriptive and very preliminary. Reliable conclusions or significantly reproducible clinical associations can rarely be made. In fact, it is somewhat disturbing to learn that 14 clinical trials are recruiting or actively testing liquid biopsy in leukemias given the studies reported in Tables 2,3 and 4. Basically, one can only agree with the authors that liquid biopsy approaches have a potential to serve as biomarkers of treatment response and MRD monitoring in the future but only pending standardization, optimization and validation of technologies for monitoring of leukemia progression or response to therapy.  It remains unclear which of the liquid biopsy components might provide optimal results as a reliable clinical biomarker. Specifically, comparative studies are needed to determine whether exosomes provide more clinically relevant information than ctDNA, given the emerging results of the presence of large, non-fragmented DNA in the vesicle lumens.  Having assembled and examined large numbers of completed studies in the Tables, the authors should strongly and plainly state in Discussion that time is NOT yet ripe for clinical application of liquid biopsy to monitoring leukemias. Fragmented preliminary data are interesting but not by any means convincing that we have a clinically reliable replacement for bone marrow biopsy.  

Author Response

We are verty thankful to the reviewer for his/her most valuable comments.

Comments 1: "Having assembled and examined large numbers of completed studies in the Tables, the authors should strongly and plainly state in Discussion that time is NOT yet ripe for clinical application of liquid biopsy to monitoring leukemias.

Response 1: We agree with this observation and for the sake of clarity we enphasized the conclusions, adding the following statement: 

"However, before a complete integration of liquid biopsy into clinical practice, key challenges remain, including standardizing cfDNA extraction protocols, enhancing the sensitivity of NGS and PCR-based assays, and achieving widespread validation through clinical trials. Addressing these challenges will be essential for the successful integration of liquid biopsy into routine hematological clinical practice."

Reviewer 3 Report

Comments and Suggestions for Authors

The review „Advancing Leukemia Management Through Liquid Biopsy: Insights into Biomarkers and Clinical Utility“ from Hollanda and colleagues provides a comprehensive overview of the current state of liquid biopsies in AML, ALL, CML and CLL.

The authors systematically present different types of analytes and methodological approaches and critically discuss the clinical contexts in which these technologies have been tested for relevance. This structure offers readers a well-rounded understanding of the field.

A major strength of the review is its organization according to leukemia subtype, which is a sensible and reader-friendly decision given the heterogeneity of these diseases. The authors illustrate their points with numerous examples, enhancing the practical value of the review. Several useful tables are included, although their ordering could be made clearer to improve navigation through the content.

The text is rich in abbreviations, which is understandable given the technical scope of the topic. However, for readers less familiar with the field, it would be helpful to reintroduce full terms at the beginning of new sections or paragraphs— as successfully applied in the discussion (e.g., line 389). This small adjustment could significantly improve readability and accessibility for a broader audience. In addition, the different leukemia enteties should be included in the abstract or keywords.

Overall, this is a valuable and informative review, which will serve both experts and those new to the field as a solid reference on the evolving role of liquid biopsies in leukemia.

Author Response

Comments 1:

"(...) for readers less familiar with the field, it would be helpful to reintroduce full terms at the beginning of new sections or paragraphs— as successfully applied in the discussion (e.g., line 389). This small adjustment could significantly improve readability and accessibility for a broader audience."

Response 1:

We are very thankful to the reviewer comments and inputs.

We agree that a considerable number of abbreviations has been used throughout the text, as would be expected in such a technical manuscript. We have carefully checked all abbreviations and substituted them for their full version whenever possible, and whenever it is used for the first time in a particular section of the text. 

Nonetheless, for the sake of clarity, organization and space usage, we would like to ask the editor whether it would be possible to have an abbreviation list as a footnote or sidenote on the front page of the publication. By doing so, the reader would have a point of reference to go to whenever they felt the need. If this suggested footnote does not meet the formatting criteria of the journal, we believe the changes already in place should be enough.

If this suggestion is accepted, we have already provided an abbreviations list below.

 ALL: acute lymphoid leukemia;

AML: acute myeloid leukemia;

CfDNA: cell-free DNA;

ctDNA: circulating tumor DNA

CLL: chronic lymphoid leukemia;

CML: chronic myeloid leukemia;

DdPCR: Digital dropplet polimerase chain reaction

ELISA: enzyme-linked immunosorbent assay

EVs: extracellular vesicles

HSCT: Hematopoietic stem cell transplantation

MiRNA: microRNA;

MRD: Minimal residual disease

NGS: New generation sequencing

SPE-CE-MS: solid-phase extraction capillary electrophoresis−mass spectrometry;

sWGS: shallow whole genome sequencing;

UMI, unique molecular identifier.

WES: Whole exome sequencing

WGS: Whole genome sequencing

Comments 2: 

"In addition, the different leukemia enteties should be included in the abstract or keywords."

Thank you for the suggestion. We have included the different leukemias in the keywords.